# Deubiquitinase Usp12 functions noncatalytically to induce autophagy and confer neuroprotection in models of Huntington's disease

Rebecca Aron[1,2,10], Pasquale Pellegrini[1], Edward W. Green[3,4], Daniel C. Maddison[3], Kwadwo Opoku-Nsiah[5], Ana Osório Oliveira[1], Jinny S. Wong[1], Aaron C. Daub[1], Flaviano Giorgini[3], Paul Muchowski[1] & Steven Finkbeiner[1,2,6,7,8,9]

Huntington's disease is a progressive neurodegenerative disorder caused by polyglutamine-expanded mutant huntingtin (mHTT). Here, we show that the deubiquitinase Usp12 rescues mHTT-mediated neurodegeneration in Huntington's disease rodent and patient-derived human neurons, and in *Drosophila*. The neuroprotective role of Usp12 may be specific amongst related deubiquitinases, as the closely related homolog Usp46 does not suppress mHTT-mediated toxicity. Mechanistically, we identify Usp12 as a potent inducer of neuronal autophagy. Usp12 overexpression accelerates autophagic flux and induces an approximately sixfold increase in autophagic structures as determined by ultrastructural analyses, while suppression of endogenous Usp12 slows autophagy. Surprisingly, the catalytic activity of Usp12 is not required to protect against neurodegeneration or induce autophagy. These findings identify the deubiquitinase Usp12 as a regulator of neuronal proteostasis and mHTT-mediated neurodegeneration.

[1] Center for Systems and Therapeutics & Taube/Koret Center for Neurodegenerative Disease, Gladstone Institutes, 1650 Owens St., San Francisco, CA 94158, USA. [2] Taube-Koret Center for Neurodegenerative Disease, San Francisco, CA 94158, USA. [3] Department of Genetics and Genome Biology, College of Life Sciences, University of Leicester, Adrian Building, University Road, Leicester LE1 7RH, UK. [4] DKFZ, Heidelberg 69120, Germany. [5] Graduate Program in Chemistry and Chemical Biology, Department of Pharmaceutical Chemistry, University of California—San Francisco, San Francisco, CA 94158, USA. [6] Graduate Program in Biomedical Sciences, University of California—San Francisco, San Francisco, CA 94158, USA. [7] Graduate Program in Neuroscience, University of California—San Francisco, San Francisco, CA 94158, USA. [8] Graduate Program in Medical Scientist Training Program, University of California—San Francisco, San Francisco, CA 94158, USA. [9] Department of Neurology and Physiology, University of California—San Francisco, San Francisco, CA 94158, USA. [10]Present address: Yumanity Therapeutics, Cambridge, MA 02139, USA. Correspondence and requests for materials should be addressed to S.F. (email: sfinkbeiner@gladstone.ucsf.edu)

everal neurodegenerative disorders, such as Huntington's disease (HD), are characterized by abnormal accumulation of toxic protein species and result in a number of deleterious effects, including dysregulation of protein homeostasis[1,2]. No disease-modifying therapies are available for any neurodegenerative disease, and as their incidence is rising rapidly in aging populations throughout the world, identifying potential therapeutic targets is of critical importance.

Although HD is known to be caused by an abnormal expansion of CAG trinucleotide repeats in the huntingtin (*HTT*) gene, phenotypic variation in HD patients with the same *HTT* mutation suggests that additional factors, both genetic and environmental, influence disease severity and age of onset[3]. Unbiased genetic approaches in a number of model systems have identified potential genetic modifiers of mutant HTT (mHTT) toxicity, but their pathogenic significance and potential as therapeutic targets remain largely unexplored[4–11].

mHTT is subject to post-translational modifications, such as ubiquitination and phosphorylation[12,13], which have important implications for its processing, stability, and localization, that subsequently influence its effect on cellular toxicity. Given the potential role of ubiquitin signaling and protein homeostasis in HD pathobiology, we sought to identify genetic modifiers of HD that are involved in these pathways. We investigated the human ortholog of a potential genetic modifier originally identified in a yeast genome-wide screen for loss-of-function enhancers of mHTT-mediated toxicity, the deubiquitinase *UBP13*[4].

*Usp12*, the eukaryotic ortholog of *UBP13*, belongs to the ubiquitin-specific protease (USP) family of deubiquitinases that are critical for ubiquitin signaling and protein quality control. USPs function as cysteine proteases to trim or remove ubiquitin conjugates from selected substrates. Deubiquitination is important for ubiquitin signaling-mediated processes, such as regulation of protein levels or activity. Similar to other USP family members, Usp12 contains the conserved catalytic cysteine/histidine domains and, so far, has only been identified as a deubiquitinase for the histones H2A/H2B, Notch, PH domain leucine-rich repeat protein phosphatase 1, and androgen receptor[14]. USPs are emerging as potential therapeutic targets for a number of human disorders in which ubiquitin-protein homeostasis pathways are disrupted, particularly cancers[15–17]. Due to their abundance and activity in the nervous system, USPs may also represent an emerging class of therapeutic targets for neurodegeneration[18].

Here, we demonstrate that Usp12 modifies mHTT-mediated toxicity in primary rodent and human patient-induced pluripotent stem cell (iPSC)-derived neurons and in vivo in a *Drosophila* model of HD. We find that Usp12 does not require its catalytic activity for this neuroprotective effect. Moreover, we reveal a non-catalytic function of Usp12 as a regulator of the cellular degradation pathway, autophagy. Autophagy induction in primary neurons and other HD models protects against HD-mediated neurotoxicity[19,20]. We propose a mechanism by which Usp12 suppresses HD-related neurodegeneration via induction of autophagy. Our findings identify ubiquitin-independent functions of Usp12 that establish it as a critical regulator of neuronal proteostasis.

## Results

### Usp12 modulates mHTT toxicity in primary neurons and flies.

We first examined the ability of Usp12 to modify mHTT toxicity in rodent primary neuron models of HD. For these studies, we used a physiologically expressed fragment of HTT, N586, which contains the first 586 amino acids of HTT, to model HD toxicity[21,22]. Our model uses a non-pathogenic form of HTT with a 17-polyglutamine-repeat expansion (Q17), or a pathogenic form with a 138-polyglutamine-repeat expansion (Q138) that is toxic to neurons. We previously described a very sensitive method to longitudinally monitor neurotoxicity in individual neurons by robotic microscopy and analysis programs[23]. This technique, coupled with statistical methods such as Cox proportional hazards analysis, allows us to track the survival of neurons and to determine the contribution of different variables to neuronal survival, such as expression level, localization, and aggregation of toxic or modifier proteins[23]. We used this system to monitor the effect of genetic depletion or overexpression of Usp12 on the survival of neurons expressing mHTT.

As the Usp12 ortholog was identified as a knockout enhancer of mHTT in the yeast HD model, we sought to determine if Usp12 was also a loss-of-function enhancer in our primary neuronal cell model of HD with the survival assay described above. Rat primary neurons were co-transfected with green fluorescent protein (GFP)-tagged HTT-N586$^{Q17}$ or -N586$^{Q138}$ and non-targeting short hairpin RNA (nt-shRNA) or shRNA targeting endogenous rat *Usp12* (*Usp12*-shRNA$^r$) (r designates rat-specific targeting sequence), and a red fluorescent protein (RFP) expression construct. The *Usp12*-shRNA$^r$ targeting construct was validated in rat cell lines—PC12 cells and C6 cells, and in rat primary neurons; Usp12 levels were determined to be reduced by ~30% at both mRNA and protein levels (Supplementary Fig. 1a-c). RFP expression was used in this assay as a survival marker: the loss of RFP fluorescence or the loss of distinctive neuronal morphology and the disruption of plasma membrane were used as an indicator of cell death[23].

Neurons transfected with these constructs were tracked at a single-cell level by robotic microscopy at approximately 24-h intervals for the indicated times (Fig. 1a). Neuronal cell survival for each cohort was analyzed by Cox proportional hazards modeling to determine the effect of Usp12 knockdown on mHTT toxicity. Neurons expressing mHTT-N586$^{Q138}$ had a greater risk of death than neurons expressing the nonpathogenic form, HTT-N586$^{Q17}$, indicating polyglutamine expansion-dependent toxicity (Fig. 1b, Table 1). *Usp12*-shRNA$^r$ significantly exacerbated the risk of death of neurons expressing toxic mHTT-N586$^{Q138}$ relative to a nt-shRNA (Fig. 1b, Table 1). These results identify Usp12 as a loss-of-function enhancer of mHTT-mediated toxicity in primary neurons, indicating that the genetic relationship between mHTT toxicity and the Usp12 ortholog is conserved from yeast to mammals, and more importantly, in an HD relevant cell type, neurons.

The exacerbation of mHTT-mediated toxicity by Usp12 depletion in neurons suggests that endogenous Usp12 has a protective role in pathways associated with mHTT toxicity. To determine if upregulating Usp12 is neuroprotective against mHTT-mediated toxicity, we overexpressed Usp12 in primary neurons expressing the mHTT constructs as above. Overexpressing human Usp12 by 2.5-fold reduced the risk of death associated with mHTT-N586$^{Q138}$ (Fig. 1c, Supplementary Fig. 1c, Table 1) but did not affect the survival of neurons expressing HTT-N586$^{Q17}$. Thus, Usp12 does not promote neuronal viability generally but protects neurons specifically against HTT-induced polyglutamine expansion-dependent toxicity. These results suggest that Usp12 is a potent and dose-dependent modulator of mHTT-dependent neurodegeneration. The specificity of Usp12 effects on mHTT-mediated toxicity was further confirmed with an siRNA that targets endogenous rat Usp12 at a distinct mRNA sequence compared to the shRNA described above (Supplementary Fig. 1d). Human Usp12 overexpressed in our studies is resistant to knockdown by *Usp12*-shRNA$^r$ and -siRNA$^r$ (Supplementary Fig. 1e). Usp12 knockdown with *Usp12*-siRNA$^r$ did not affect HTT-N586$^{Q17}$ survival (Fig. 1d, Supplementary Table 1) but increased mHTT-N586$^{Q138}$ toxicity (Fig. 1e, Supplementary

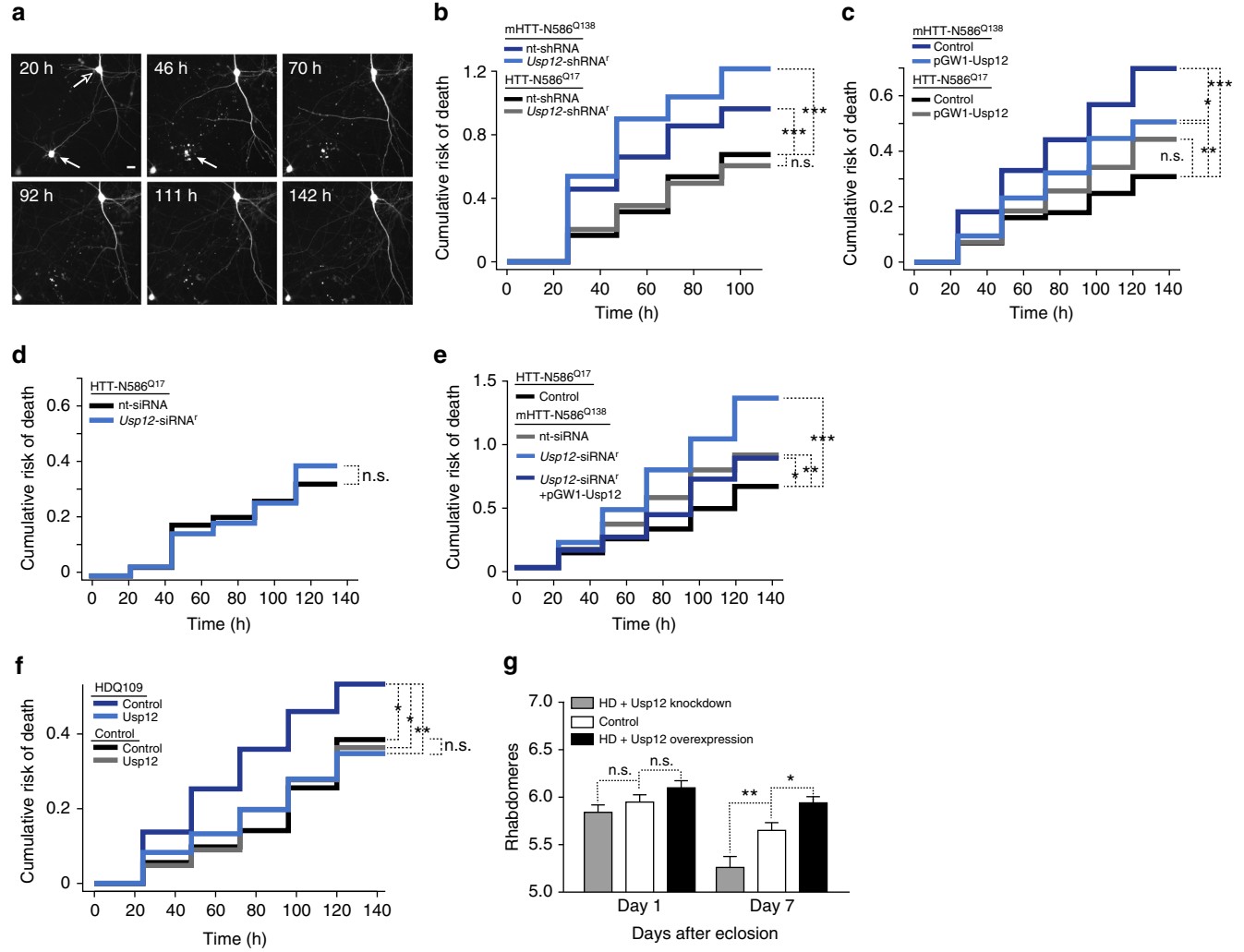

**Fig. 1** Usp12 is a modifier of mHTT toxicity in vitro and in vivo. **a** Representative images of primary neurons expressing RFP captured at the indicated time intervals after transfection. Expression of RFP is used as a morphological marker to track the survival of neurons by a robotic microscope. Examples of neurons that survived the entire time course of observation (empty arrowhead) or died by 46 h post transfection (filled arrowhead) are shown (scale bar = 20 μm). **b–e** Cumulative risk of death plots of primary rodent cortical neurons. The cumulative risk of neuron death is plotted against time (h) for each group of neurons transfected with the indicated plasmids. Cox proportional hazards analysis was used to estimate the relative risk of death or hazard ratio (HR), and the $p$ value was determined with the log rank test (*$p < 0.05$, **$p < 0.01$, ***$p < 0.001$, n.s. not significant) (for statistical information, see Supplementary Tables 1, 2, and 3). **f** Survival analysis of human neurons differentiated from patient-derived iPSCs (unaffected individual, control; HD patient, HDQ109) transfected with MAP2-RFP (morphology marker), and Usp12 or control vectors (statistical information is summarized in Supplementary Table 3). **g** Quantification of average rhabdomeres per ommatidium at day 1 and day 7 after eclosion. Two-way ANOVA: genotype effects, $p < 0.0001$; age effects, $p < 0.0001$, $n = 8$–10 flies per genotype. Multiple comparisons analysis: day 1 genotype analysis for control vs. Usp12 knockdown ($p = 0.7020$) and control vs. Usp12 overexpression ($p = 0.3589$); day 7 genotype analysis for control vs. Usp12 knockdown ($p = 0.0041$) and control vs. Usp12 overexpression ($p = 0.0270$). n.s. not significant; *$p < 0.05$; **$p < 0.01$. Two-way ANOVA with Bonferroni's multiple comparisons post hoc test. Error bars represent s.e.m.

Table 2). The specificity of the increased toxicity observed upon Usp12 knockdown was confirmed by overexpressing human Usp12 in neurons. Usp12 rescued the toxicity mediated by the *Usp12*-siRNA[r] (Fig. 1e, Supplementary Table 2).

We next sought to further establish the physiological relevance of Usp12 as a modifier of mHTT using human neurons from HD patients that express full-length mHTT under its endogenous promoter. Neurons differentiated from HD patient-derived iPSCs exhibit several features reminiscent of HD pathobiology, such as changes in gene/protein expression, the cytoskeleton, adhesion, electrophysiological properties, energy metabolism, and survival[24]. For this assay, we differentiated neuronal cells from an HD patient with 109 polyglutamine repeats (HDQ109) and an unaffected individual with 20 polyglutamine repeats (control).

Usp12 overexpression significantly decreased the risk of death of HDQ109 neurons, but did not affect the survival of control neurons (Fig. 1f, Supplementary Table 3).

To assess the physiological relevance of our findings from cell models in an in vivo system, we examined the effect of Usp12 on neurodegeneration in an extensively used *Drosophila* model of HD[25]. Expression of an expanded polyglutamine exon 1 fragment (mHTT-ex1Q93) in *Drosophila* leads to progressive loss of photoreceptor neurons[25], as well as other disease-relevant phenotypes. Using this feature of HD model flies to monitor neurodegeneration, we determined the effect of Usp12 knockdown or Usp12 overexpression on the progressive loss of photoreceptor neurons (Fig. 1g, Supplementary Fig. 1f). Usp12 knockdown significantly increased, and Usp12 overexpression

**Table 1 Cox proportional hazards analysis of the effects of Usp12 knockdown or overexpression on survival of rodent primary neurons**

| | HR | 95% CI | *P* value | *n* |
|---|---|---|---|---|
| **Usp12 knockdown** | | | | |
| HTT-N568$^{Q17}$ + nt-shRNA | Reference | – | – | 222 |
| HTT-N568$^{Q17}$ + *Usp12*-shRNA$^r$ | 0.94 | 0.69–1.27 | 0.67050 | 141 |
| mHTT-N586$^{Q138}$ + nt-shRNA | 1.58 | 1.23–2.05 | 0.00042 | 207 |
| mHTT-N586$^{Q138}$ + *Usp12*-shRNA$^r$ | 2.00 | 1.56–2.57 | 4.27e-08 | 209 |
| **Usp12 overexpression** | | | | |
| HTT-N568$^{Q17}$ + control vector | Reference | – | – | 196 |
| HTT-N568$^{Q17}$ + Usp12 | 1.41 | 0.98–2.03 | 0.06154 | 190 |
| mHTT-N586$^{Q138}$ + control vector | 2.27 | 1.63–3.18 | 1.47e-06 | 199 |
| mHTT-N586$^{Q138}$ + Usp12 | 1.64 | 1.15–2.34 | 0.00604 | 189 |
| HTT-N568$^{Q17}$ + control vector | 0.44 | 0.31–0.61 | 1.47e-06 | 196 |
| HTT-N568$^{Q17}$ + Usp12 | 0.62 | 0.45–0.84 | 0.00229 | 190 |
| mHTT-N586$^{Q138}$ + control vector | Reference | – | – | 199 |
| mHTT-N586$^{Q138}$ + Usp12 | 0.72 | 0.53–0.97 | 0.03165 | 189 |

Reference = reference group for statistical analysis
HR hazard ratio, *n* number of neurons, CI confidence interval

decreased, mHTT-associated loss of photoreceptor neurons and neurodegeneration. As ubiquitous or pan-neuronal knockdown of *Drosophila* Usp12 results in no identifiable phenotypes[26], the exacerbated loss of photoreceptor neurons in the HD transgenic flies can be attributed to a specific effect of Usp12 on mHTT-associated neurodegeneration, rather than an adverse effect of Usp12 loss of function in neuronal cells.

**Usp12 is a specific modifier of mHTT toxicity.** Identification of Usp12 as a modifier of HD suggested that Usp12 could rescue neurodegeneration in related disorders, such as amyotrophic lateral sclerosis (ALS) and Parkinson's disease (PD) that are characterized by proteotoxic stress. Like HD, these neurodegenerative diseases feature accumulations of toxic misfolded proteins and the appearance of ubiquitin-positive neuronal inclusions.

We tested the effect of Usp12 on neuronal cell survival in primary neuron models of ALS and PD. The toxic neurodegenerative disease-associated proteins, TDP-43, an ALS-causing mutant of TDP-43, TDP-43$^{A315T}$, and α-synuclein show overt toxicity and lead to neuron death when overexpressed in primary neurons[27,28]. To determine if Usp12 could modify toxicity of these disease proteins, we tracked the survival of neurons co-transfected with Usp12 and either TDP-43 wild type (WT), TDP-43$^{A315T}$, or α-synuclein expression constructs. In agreement with previous results, overexpression of TDP-43 and α-synuclein in primary neurons significantly increased risk of death (Fig. 2a–c, Supplementary Table 4). Co-expression with Usp12 did not reduce the risk of neuron death in the TDP-43 or the α-synuclein disease models (Fig. 2, Supplementary Table 4) as it did in our mHTT models (Fig. 1), indicating that Usp12 is not suppressing neurotoxicity generically.

**Usp46 does not rescue mHTT-mediated neuronal death.** To better understand the potential specificity of Usp12 in HD pathobiology, we compared the effects of Usp12 on mHTT-mediated toxicity with those of similar related enzymes of the USP family. The most closely related human homolog to Usp12 is Usp46 (93% protein sequence similarity, with identical active-site domain sequences; Fig. 3a). Using the robotic microscopy survival assay described above, we tracked the survival of neurons overexpressing Usp46 and mHTT. When transfected at a similar gene dose as that at which Usp12 rescues mHTT

toxicity (Fig. 3b), Usp46 did not block degeneration of mHTT-expressing neurons (Fig. 3c, e, Supplementary Table 5). However, Usp46 overexpression appeared to increase the risk of death in neurons co-transfected with a non-pathogenic form of HTT, HTT-N586$^{Q17}$ (Fig. 3d, Supplementary Table 5). This effect depended on the active-site cysteine, C44, as overexpression of Usp46-C44S did not increase death in cells expressing HTT-N586$^{Q17}$ (Fig. 3d). Usp46-C44S did not reduce the toxicity of mHTT-N586$^{Q138}$, indicating that the toxicity seen here with mHTT depends specifically on expression of mHTT-N586$^{Q138}$ (Fig. 3e). To demonstrate that the reduction in toxicity in neurons expressing Usp46-C44S was not due to the C44S mutation effecting protein expression or stability, we tested the expression of Usp46 and Usp46-C44S in PC12 cells. The Usp46-C44S construct produces protein as efficiently and stably as the WT Usp46 construct (Fig. 3g). No overt toxicity was observed when equivalent amounts of Usp12 plasmid were transfected into control neurons (Fig. 3f). Thus, despite the similarity in protein sequence and conservation of active sites of Usp12 and Usp46, they have unique neurobiological functions. These results further suggest that Usp12 has a function distinct from related USPs, in suppressing mHTT-mediated toxicity.

**Catalytically inactive Usp12 suppresses mHTT toxicity.** The inability of Usp46 to suppress mHTT neurotoxicity, despite having identical catalytic domain sequences to Usp12, suggested that functional specificity for either of these enzymes resides outside of these domains, and that catalytic activity was not required for this function of Usp12. Therefore, we tested the effect of modulating Usp12 deubiquitinase activity on mHTT toxicity. Usp12 contains a conserved active-site cysteine residue (C48) (Fig. 4a). C48 is required for Usp12 deubiquitinating activity in vitro, as Usp12 with a replacement of this residue by serine (Usp12-C48S) lacks enzymatic activity[29]. To test the requirement of Usp12 deubiquitinating activity in modulating mHTT toxicity, we examined the effect of Usp12-C48S on mHTT toxicity in our robotic microscopy survival assay. Despite the mutation, Usp12-C48S retained the ability to rescue mHTT-mediated toxicity in primary neurons, indicating that an active-site cysteine is not required for rescue of mHTT toxicity (Fig. 4b, Table 2).

In order to rule out the possibility that residual protease activity in the context of the cellular environment contributes to

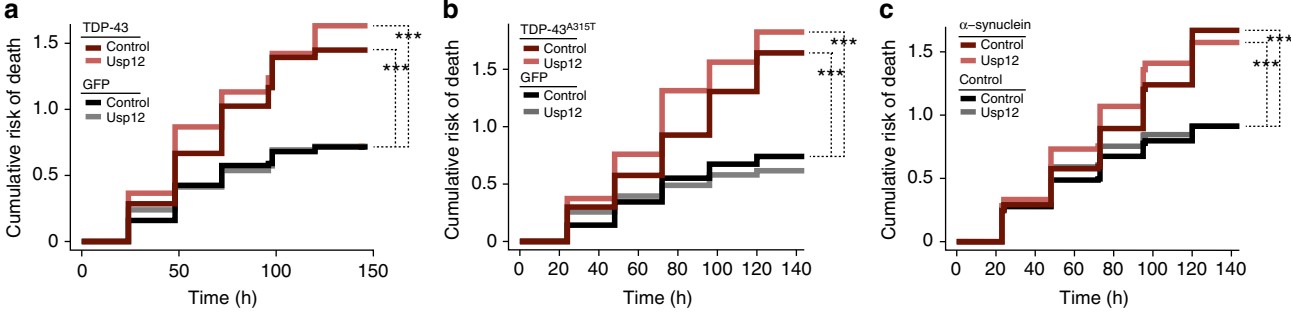

**Fig. 2** Usp12 overexpression is not a generic suppressor of neurotoxicity. **a–c** Cumulative risk of death plots for primary neurons co-transfected with control, TDP-43 (**a**), TDP-43$^{A315T}$ (**b**), α-synuclein (**c**), and Usp12 or empty vector control plasmids. Cox proportional hazards analysis was used to estimate the relative risk of death, or hazard ratio (HR), and the p value was determined with the log rank test (***p < 0.001) (Supplementary Table 4)

Usp12 suppression of mHTT-mediated toxicity, we mutated C50, a cysteine residue adjacent to C48, to serine and determined if it affected the ability of Usp12 to suppress mHTT toxicity. Usp12-C50S and Usp12 with both C48S and C50S mutations (Usp12-DBLCS) suppressed mHTT toxicity as efficiently as WT Usp12 and Usp12-C48S (Fig. 4c, Table 2). Similar to WT Usp12, the overexpression of Usp12-C48S rescued the toxicity of Usp12 knockdown (Supplementary Fig. 2a, b, Supplementary Table 2). Expression and stability of WT Usp12 and mutants were comparable when overproduced in human embryonic kidney (HEK) cells (Fig. 4d). These results suggest that Usp12 does not require C48 for its function in suppressing mHTT toxicity and, furthermore, that an adjacent cysteine residue does not compensate for the loss of C48 to suppress mHTT toxicity. If the deubiquitinating activity of Usp12 is not required for its ability to mitigate mHTT toxicity, we would predict that Wdr20 and Wdr48, two essential co-factors of Usp12[29,30], would also not be required for Usp12-mediated neuroprotection (Fig. 4e). Indeed, we found that Usp12-C48S reduces mHTT toxicity despite the knockdown of Wdr48 and/or Wdr20 (Fig. 4f, g, Supplementary Table 6).

To determine if the catalytic activity of Usp12 was dispensable for neuroprotection against mHTT toxicity in human neurons, we tested the effect of Usp12-C48S on survival of an HD patient line with 53 polyglutamine repeats (HDQ53)[31]. The HDQ53 line showed developmental alterations and higher risk of death than control lines[31]. Similar to results with the HDQ109 line, Usp12 significantly increased survival in HD53Q differentiated neurons to the level of control neuron survival (Figs. 1f and 4h, Table 3). Usp12-C48S showed a similar level of neuroprotection in HD53Q neurons, while not affecting survival of the control line (28 polyglutamine repeats; Fig. 4h, Table 3). As the conversion of an active-site cysteine to serine could result in generating an active serine protease[32], we also tested the effect of mutating the active-site cysteine to a less-conserved residue, alanine. Usp12-C48A rescued mHTT toxicity in our neuron model in a manner similar to Usp12-C48S (Supplementary Fig. 3).

Our results indicate that an active-site cysteine is not required for Usp12 function in neuroprotection, and Usp46, despite containing an identical catalytic domain sequence, does not rescue mHTT-mediated toxicity. These findings further support the concept that Usp12 functions by modulating mHTT-mediated neurotoxicity in a non-canonical manner.

**Usp12 does not affect inclusion body dynamics**. We next explored potential ubiquitin-independent mechanisms by which Usp12 might modulate mHTT toxicity. Controlling for other variables, the accumulation of mHTT increases, and mHTT inclusion body (IB) formation decreases the risk of neuronal death[23]. Possible mechanisms by which Usp12 could function are by modulating mHTT levels, aggregation, or IB dynamics in a ubiquitin-independent manner.

To examine the effect of Usp12 on mHTT turnover, we measured the half-life of mHTT proteins fused to the photo-convertible protein, mEos2, in our primary neuron model with an optical pulse-labeling assay. This approach is more suitable and sensitive than conventional metabolic pulse chase assays to measure the clearance of toxic, aggregation-prone proteins[33,34]. mEos2 fluoresces green, but when pulsed with 405 nm wavelength, the existing protein population is photoconverted to a red fluorescent form. The half-life of mHTT-mEos2 can be determined by measuring the decay of the photoconverted red form over time. Half-life of soluble mHTT protein was determined by analyzing the decay rate of only visibly diffuse mHTT protein. Neither Usp12 nor Usp12-C48S significantly modified either the mean half-life or distribution of half-lives of soluble mHTT (Supplementary Fig. 4).

To visualize potential co-localization of Usp12 with mHTT in neurons, we utilized GFP- or RFP-tagged versions of Usp12. Fluorescent protein-tagged Usp12 retains its function in suppressing mHTT toxicity in primary neurons, and therefore represents a functional version of Usp12 suitable for our studies here (Supplementary Fig. 5). We observed that both RFP-Usp12-WT and -C48S localize to mHTT-N586$^{Q138}$-GFP inclusions in primary neurons (Fig. 5a). We next used Cox proportional hazards analysis to determine the influence of Usp12 on the risk factors for mHTT-mediated toxicity, expression levels, and IB formation. Usp12 and Usp12-C48S did not affect these risk factors for mHTT-mediated toxicity (Supplementary Table 7). We also tested the effect of Usp12 or Usp12-C48S on the risk of IB formation, which if increased, could indicate a protective mechanism of Usp12. Our analysis indicated that neither Usp12 nor Usp12-C48S affects the risk of mHTT IB formation (Supplementary Fig. 6, Supplementary Table 8).

**Usp12 regulates autophagy in primary neurons**. Usp12 did not seem to influence mHTT levels, the propensity of neurons to form mHTT IBs, or the contribution of risk factors to mHTT-mediated toxicity and neuronal survival. Thus, we searched for an alternative connection between Usp12 and mHTT. Using the available association data for Usp12 and mHTT interactions, we mapped a protein–protein interaction network between Usp12 and mHTT, and identified the autophagy receptor Optineurin (Optn) as a potential link (Fig. 5b, Supplementary Fig. 7). A physical interaction between Optn and mHTT has been reported, and Optn interacts with Wdr20, a cofactor that directly interacts with Usp12[14,35–38]. Inhibition of autophagy results in the accumulation of autophagy receptors in punctate structures.

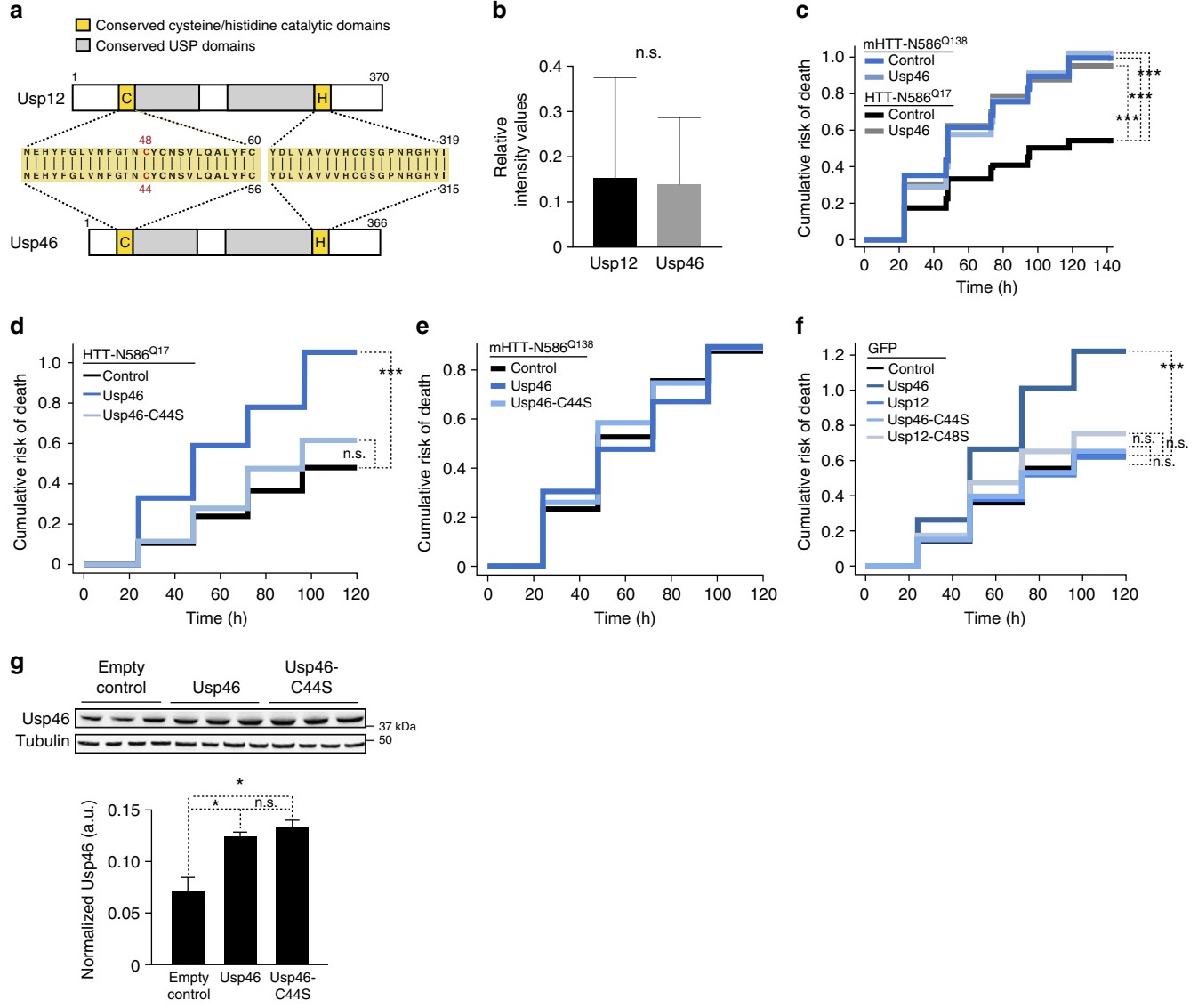

**Fig. 3** Differential effects of Usp12 and Usp46 on mHTT toxicity and neuronal survival. **a** Schematic representation of Usp12 and Usp46 protein domain structure with sequence alignment of the conserved regions at the active-site cysteine and histidine domains (highlighted in yellow). **b** Relative intensity values of primary neurons transfected with GFP-Usp12 ($n = 47$) and GFP-Usp46 ($n = 46$) plasmids normalized to RFP control. Data are representative of three independent experiments and values are expressed as mean ± s.d. $p$ Value ($p = 0.72$) was determined with unpaired $t$ test. **c–e** Cumulative hazard plots for primary neurons co-transfected with HTT-N568Q17 or mHTT-N568Q138, and Usp46 or Usp46-C44S and controls. **f** Neuronal toxicity of Usp12 and Usp46 co-transfected with GFP control. Cox proportional hazards analysis was used to estimate the relative risk of death, or hazard ratio (HR), and the $p$ value was determined with the log rank test (***$p < 0.001$) (Supplementary Table 5). **g** Expression and stability of Usp46 and Usp46-C44S are equivalent in PC12 cells transfected with these plasmids. PC12 cells co-transfected with empty vector, Usp46, or Usp46-C44S expression constructs were lysed and subjected to SDS-PAGE and immunoblotting with Usp46 antibody. Tubulin was assayed as a loading control. One-way ANOVA, multiple comparison. Normalized Usp46 intensity of the empty vector vs. Usp46 ($p = 0.0457$), empty vector vs. Usp46-C44S ($p = 0.0138$), and Usp46 vs. Usp46-C44S ($p = 0.5855$, s.e.m. are shown)

Upon inhibiting autophagy with ammonium chloride or bafilomycin A, Usp12 and Usp12-C48S co-localized with Optn in punctate structures throughout the cell (Fig. 5c, Supplementary Fig. 8a). About 60% and 40% of the Usp12 and Usp12-C48S puncta, respectively, co-localized with Optn (Supplementary Fig. 8c). Co-immunoprecipitation confirmed an interaction between Usp12 or Usp12-C48S and Optn (Fig. 5e).

To further validate the interaction of Usp12 with autophagy receptors, we examined the autophagy receptor p62, finding that both Usp12 and Usp12-C48S co-localized with p62 in punctate structures when autophagy was inhibited (Fig. 5d, Supplementary Fig. 8b), an observation that was confirmed by super-resolution

microscopy (Supplementary Fig. 9a, b). Usp12 (80%) and Usp12-C48S (50%) puncta co-localized with p62 (Supplementary Fig. 8d). An interaction between Usp12 or Usp12-C48S and p62 was also confirmed by co-immunoprecipitation (Fig. 5f). To investigate the interaction of Usp12 with endogenous autophagic receptors, we observed RFP-Usp12-expressing neurons labeled with p62 antibodies using super-resolution microscopy. Similar to previous data, we observed a subset of p62 and Optn puncta co-localized with Usp12 (Supplementary Fig. 9a, b).

To investigate a potential function of Usp12 in modulating autophagy, we used an optical pulse-labeling assay to determine the effect of overexpressing or depleting Usp12 on the half-life of

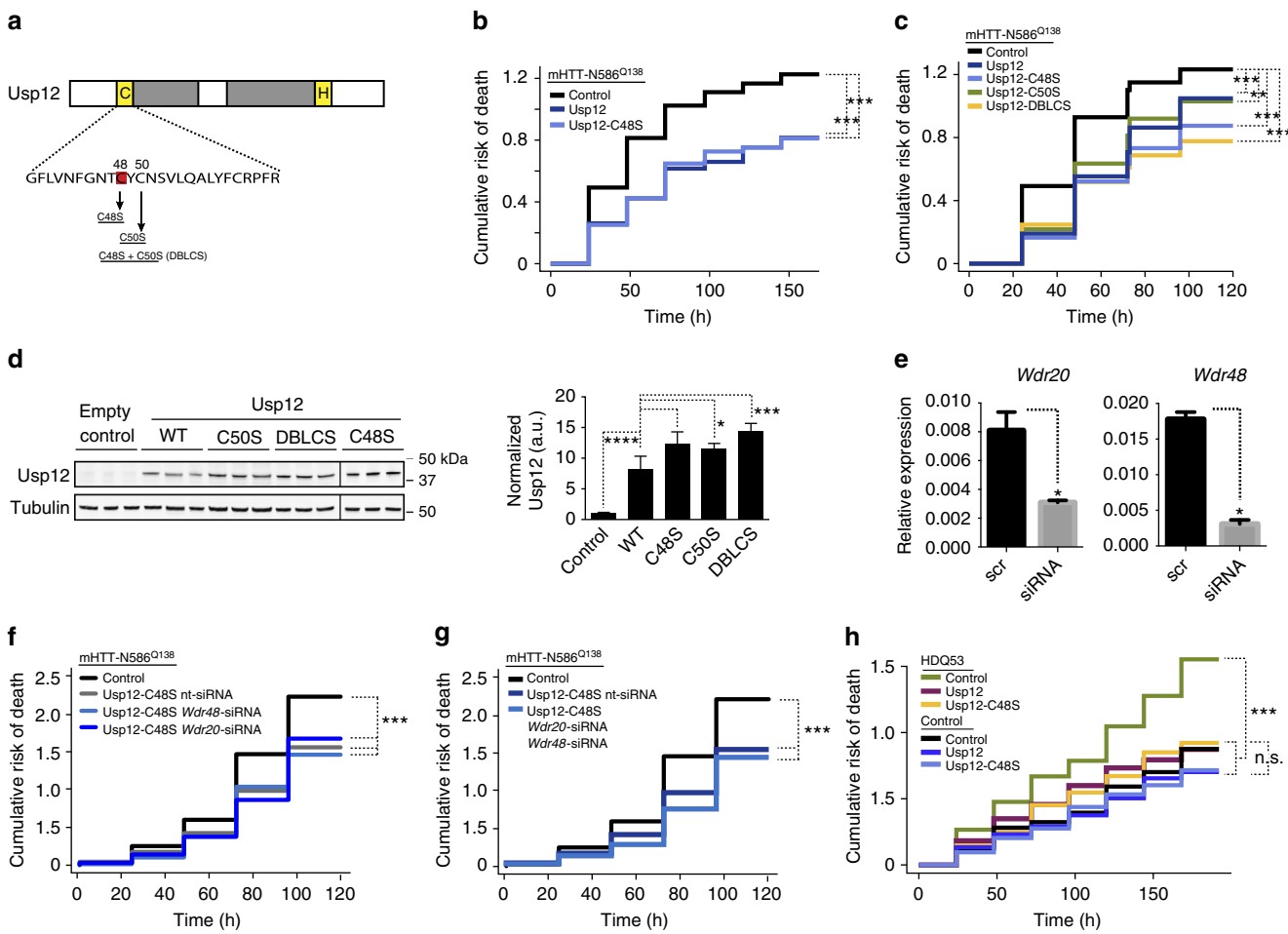

**Fig. 4** Deubiquitinating activity of Usp12 is not required for suppression of mHTT toxicity. **a** Schematic representation of Usp12 domain structure. The conserved active-site cysteine (C48) is highlighted in red. **b, c** Cumulative risk of death plots for primary neurons co-transfected with mHTT-N568$^{Q138}$ and Usp12, Usp12 mutants that lack deubiquitinating activity in vitro (Usp12-C48S), or retain activity (Usp12-C50S), Usp12 containing both mutations (Usp12-DBLCS), or empty vector control; statistical data are summarized in Table 2. **d** Expression of Usp12 and Usp12 mutants in HEK cells transfected with these plasmids. Usp12-C48S samples are from a separate gel. Tubulin was assayed as a loading control, s.e. are shown. One-way ANOVA with multiple comparisons, ****$p < 0.0001$, ***$p < 0.001$, *$p < 0.05$. **e** Relative *Wdr20* and *Wdr48* mRNA levels of a rat C6 cell line transfected with *Wdr20* and *Wdr48* siRNAs, respectively. *$p < 0.01$, s.e.m. are shown. **f, g** Cumulative risk of death plots for primary neurons co-transfected with mHTT-N568$^{Q138}$, Usp12-C48S, and siRNAs against either Wdr20 or Wdr48 (**f**) or both (**g**). HR was used to estimate the relative risk of death and results are summarized in Supplementary Table 6. **h** Survival analysis of human neurons differentiated from patient-derived iPSCs (unaffected individual, control; HD patient, HDQ53). Cox proportional hazards analysis was used to estimate the relative risk of death, or hazard ratio (HR), and the *p* value was determined with the log rank test (*$p < 0.05$, **$p < 0.01$, ***$p < 0.001$) (Table 3)

| Table 2 Cox proportional hazards analysis of the effects of Usp12 catalytic domain mutants on neuron survival | | | | |
|---|---|---|---|---|
| | **HR** | **95% CI** | **P value** | **n** |
| mHTT-N586$^{Q138}$ + control vector | Reference | – | – | 319 |
| mHTT-N586$^{Q138}$ + Usp12 | 0.71 | 0.59–0.86 | 0.51e-03 | 294 |
| mHTT-N586$^{Q138}$ + Usp12-C48S | 0.61 | 0.50–0.75 | 1.74e-06 | 283 |
| mHTT-N586$^{Q138}$ + Usp12-C50S | 0.73 | 0.60–0.89 | 0.22e-02 | 266 |
| mHTT-N586$^{Q138}$ + Usp12-DBLCS | 0.58 | 0.48–0.72 | 5.84e-08 | 324 |
| Reference = reference group for statistical analysis | | | | |
| HR hazard ratio, *n* number of neurons, CI confidence interval | | | | |

the autophagy protein LC3 in primary neurons. LC3, a mammalian homolog of yeast Atg8, incorporates into autophagosomes, and is selectivity degraded via autophagy[39,40]. The LC3 half-life reflects autophagic activity and can be used as a sensitive reporter of autophagic flux[39]. In our assay, the half-life of LC3

fused to a photoswitchable protein, Dendra2-LC3, is measured in individual, living neurons by robotic microscopy[33,34]. Dendra2-LC3 in the green channel represents both the Dendra2-LC3 existing at the time of photoconversion and the newly synthesized protein throughout the experiment. Dendra2-LC3 in the

**Table 3 Cox proportional hazards analysis of non-catalytic Usp12 effect on survival of patient-derived iPSC-differentiated neurons**

|  | HR | 95% CI | P value | n |
|---|---|---|---|---|
| Control cells + control vector | Reference | – | – | 163 |
| Control cells + Usp12 | 0.84 | 0.58–1.20 | 0.339 | 88 |
| Control cells + Usp12-C48S | 0.84 | 0.55–1.29 | 0.434 | 55 |
| HDQ53 cells + control vector | 1.79 | 1.39–2.31 | 6.13e-06 | 227 |
| HDQ53 cells + Usp12 | 1.10 | 0.83–1.46 | 0.498 | 180 |
| HDQ53 cells + Usp12-C48S | 1.10 | 0.83–1.47 | 0.499 | 161 |
| Control cells + control vector | 0.56 | 0.43–0.72 | 6.13e-06 | 163 |
| Control cells + Usp12 | 0.47 | 0.34–0.65 | 8.68e-06 | 88 |
| Control cells + Usp12-C48S | 0.47 | 0.31–0.71 | 0.000274 | 55 |
| HDQ53 cells + control vector | Reference | – | – | 227 |
| HDQ53 cells + Usp12 | 0.62 | 0.48–0.78 | 9.25e-05 | 180 |
| HDQ53 cells + Usp12-C48S | 0.62 | 0.48–0.79 | 0.000172 | 161 |

Reference = reference group for statistical analysis
HR hazard ratio, n number of neurons, CI confidence interval

red channel represents only the population existing at photoconversion (Fig. 6a). By following the decay of Dendra2-LC3 fluorescence in the red channel, we can determine the clearance rate of LC3, without interference from newly synthesized Dendra2-LC3 protein.

Usp12 and Usp12-C48S overexpression significantly decreased the mean half-life of LC3, but did not affect the half-life of the control protein Dendra2 (Fig. 6b, c). Usp12 and Usp12-C48S also significantly left-shifted the distribution of LC3 half-lives in individual neurons, but not the control protein Dendra2. Conversely, Usp12 knockdown increased the mean half-life of LC3, while also significantly right-shifting the distribution of LC3 half-lives in individual neurons (Fig. 6d). Usp12 knockdown did not affect half-life of the control protein Dendra2 (Fig. 6e).

We next examined the effects of Usp12 on autophagic structures in primary neurons by electron microscopy (EM). Neurons transduced with either Usp12 overexpression lentivirus or GFP control lentivirus were examined by EM. Neurons overexpressing Usp12 had approximately sixfold more autophagic structures, such as pre-autophagosomes, autophagosomes, and autolysosomes, than GFP lentivirus–transduced neurons (Fig. 6f, Supplementary Fig. 10).

To further confirm that the effect of Usp12 on LC3 turnover reflects a functional role of Usp12 in autophagy, we tested the effect of Usp12 on LC3 half-life in the context of genetic depletion of Atg7. Genetic depletion of Atg7 by shRNA knockdown increases Dendra2-LC3 half-life in primary neurons in an optical pulse-labeling assay similar to the one used here[33]. Consistent with these results, we showed that Usp12 no longer effects changes in Dendra2-LC3 half-life in the context of Atg7-shRNA knockdown (Fig. 7a, b), indicating that Usp12-mediated LC3 turnover in our assay reflects changes specific to an Atg7-dependent autophagy pathway of LC3 clearance.

Dysregulation of autophagy is linked to a number of neurodegenerative disorders, including HD[41]. Usp12 induction of autophagy is a potential mechanism by which Usp12 confers neuroprotection in HD models. To investigate the relationship between Usp12 function in autophagy and neuroprotection, we tested the ability of Usp12 to modulate mHTT toxicity in neurons repressed for autophagy by knockdown of Atg7, a regulator of autophagy. Usp12 nor Usp12-C48S suppressed mHTT-N586$^{Q138}$

neurotoxicity during Atg7 knockdown (Fig. 7c, Supplementary Table 9). By contrast, Usp12 or Usp12-C48S suppressed mHTT-N586$^{Q138}$-mediated neurotoxicity when co-expressed with non-targeting-shRNA, indicating that the neuroprotective effects of Usp12 or Usp12-C48S require a functioning autophagic clearance pathway (Figs. 7c and 1c). The dependence of Usp12-mediated neuroprotection on Atg7-dependent autophagy supports our model, whereby Usp12 modulates mHTT toxicity via its function in autophagy induction.

## Discussion

There are no effective therapies for HD. Studies exploring potential genetic disease-modifying agents have the potential to lead to the identification of specific molecular targets, and further our understanding of relevant molecular pathways for therapeutic intervention. In our studies, we examined the candidate modifier Usp12 for its effect on mHTT toxicity. Given the wide debate on reliable pre-clinical models that recapitulate HD, here we validated our findings in multiple complementary models to ensure relevance to human disease. We identified Usp12 as a modifier of mHTT-associated neurodegeneration in rodent primary neurons, human HD patient-derived iPSC-differentiated neurons and in vivo using a *Drosophila* model of HD. Investigation into the neuroprotective mechanism of Usp12 revealed that Usp12 is a component of the autophagic machinery and a regulator of autophagic flux in neurons.

Initially, we hypothesized that mHTT was a substrate of Usp12 deubiquitinating activity, which could be a potential mechanism for regulating stability, localization, or other ubiquitin-related signaling activity of ubiquitin-conjugated mHTT. Interestingly, we found that Usp12 does not require its deubiquitinating activity to suppress mHTT neuronal toxicity. Although still thought to be relatively uncommon, non-canonical functions of other deubiquitinases have been reported. The deubiquitinase OTUB1 functions non-catalytically in regulation of DNA repair, and the Usp14 ortholog in yeast, Ubp6, functions non-catalytically to delay proteasome-dependent protein degradation[42,43]. Our studies identifying non-catalytic functions of Usp12 support the idea that deubiquitinases may generally function as multimodal proteins. Identification of potential non-catalytic functions of deubiquitinases could be of significance regarding effects or consequences of catalytic site inhibition, which is becoming an increasingly studied therapeutic strategy[44].

The most closely related human homolog of Usp12, Usp46, did not function in a manner similar to Usp12 in suppressing mHTT toxicity when transfected into neurons at the same gene dosage. Expression of Usp46, in contrast to Usp12, is neurotoxic on its own, as we observed an increased risk of death of neurons overexpressing Usp46. The toxic effect of Usp46 depended on its deubiquitinating activity as expression of the catalytically inactive mutant of Usp46 was not toxic to neurons. These results suggest that Usp12 and Usp46 have distinct neurobiological functions, despite sharing >90% amino-acid sequence similarity and identical catalytic domain sequences. Furthermore, these results suggest that the functional specificity of Usp12 in suppressing mHTT toxicity is conferred by sequences that reside outside the conserved catalytic domains, and further support the idea that Usp12 has distinct, non-catalytic functions among related deubiquitinases. Future studies characterizing the regulation of expression, localization, and activity of these proteins in neuronal cells will lead to a better understanding of the mechanisms conferring their functional specificity.

A potential coping mechanism that neurons may utilize in response to expression of toxic mHTT is the acute formation of IBs. Indeed, mHTT IB formation is a predictor of neuronal

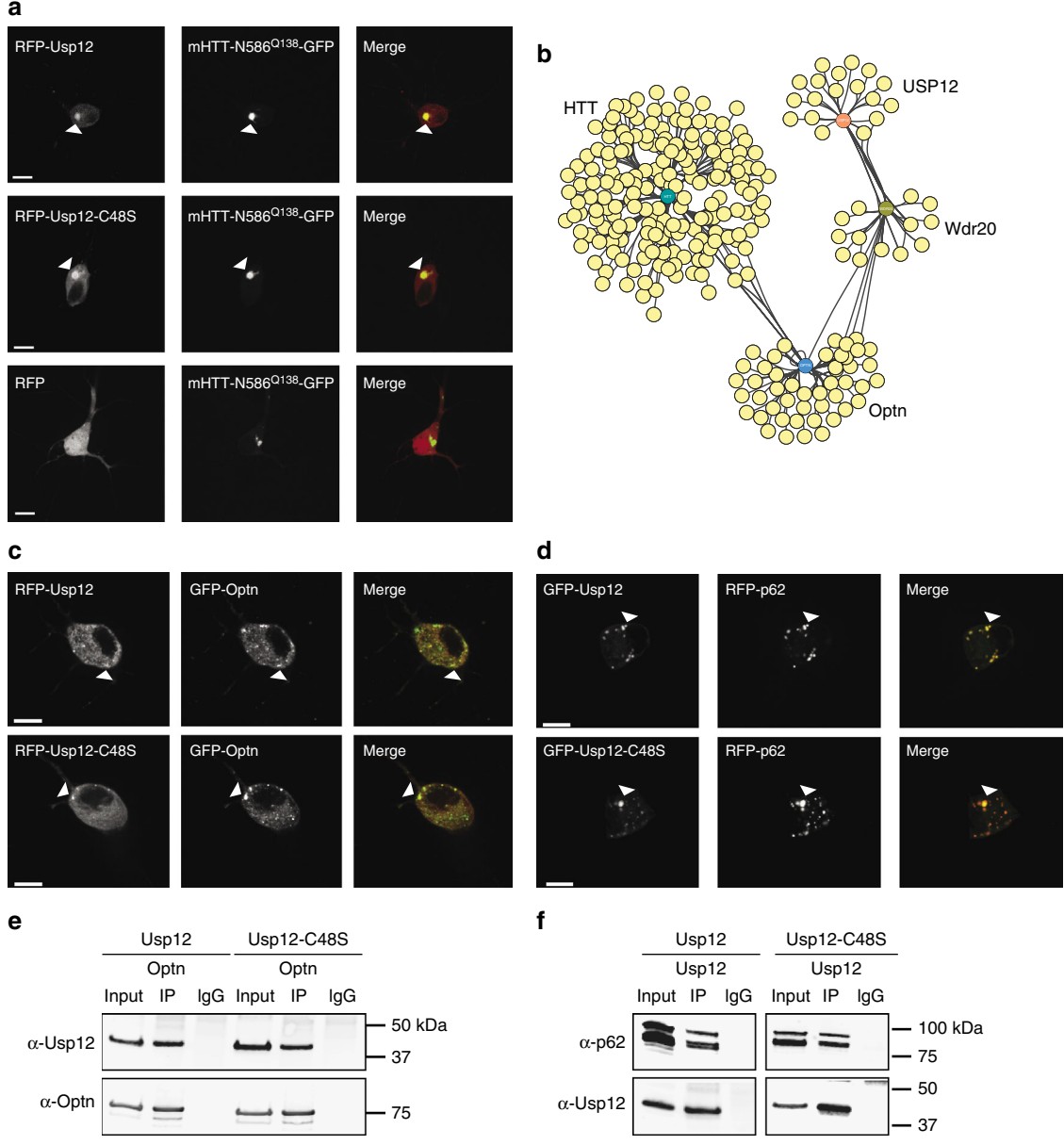

**Fig. 5** Usp12 localizes to mHTT-N586$^{Q138}$ inclusion bodies and autophagy receptors. **a** Representative images of RFP-Usp12 and mHTT-N568$^{Q138}$-GFP expression and localization in primary neurons. Images were captured in live cells approximately 48 h after transfection. Co-localization of Usp12 puncta and IBs is indicated (arrowheads). **b** Map of protein interaction networks between HTT, Usp12, Wdr20, and Optn, based on data collection in BioGRID. Detailed map is in Supplementary Fig. 7. **c** Representative images of GFP- or RFP-Usp12 and -Usp12-C48S neurons co-transfected with either GFP-Optn or **d** RFP-p62 and treated with NH$_4$Cl. Co-localization of Usp12 and Optineurin or p62 puncta is indicated (arrowheads), scale bar = 10 μm. **e**, **f** Immunoprecipitation of Optn (**e**) or Usp12 (**f**), followed by immunoblotting with anti-Usp12, anti-Optn, or anti-p62 antibodies in HEK cells transfected with Usp12, Usp12-C48S, Optn, or RFP-p62 as indicated

survival. Neurons that form IBs survive longer, possibly by sequestering toxic mHTT species into a relatively more inert state[23]. We observed that Usp12 co-localizes to mHTT IBs in primary neurons; however, we saw no effect of Usp12 on the proportion of cells that formed IBs. Usp12 may be functioning to alter IB dynamics, such as stability, content, or localization, which a more in-depth biochemical analysis could reveal. As autophagy is the primary clearance pathway for mHTT IBs[20], the Usp12/mHTT IB interaction might be directly relevant to the function of Usp12 that we discovered in modulating autophagy. It is conceivable that Usp12 has additional yet unknown autophagy substrates or receptors that are necessary for the selective clearance of subspecies of mHTT IBs.

Stimulating the ubiquitin proteasome system (UPS) or autophagic clearance pathways is also a protective coping mechanism in response to the expression and accumulation of toxic mHTT protein. mHTT containing a pathogenic polyglutamine length, such as Q46, is preferentially sensitive to autophagic rather than UPS-mediated clearance[34]. We reported development of a sensitive assay for measuring autophagic flux in single neurons by optical pulse labeling[33,34]. This assay allows us to measure the clearance rate of the autophagy substrate LC3. As presented here, neurons expressing Usp12 cleared LC3 faster than control neurons. Conversely, neurons in which Usp12 expression was knocked-down cleared LC3 more slowly than control neurons. Using the clearance rate of LC3 as a sensitive indicator of

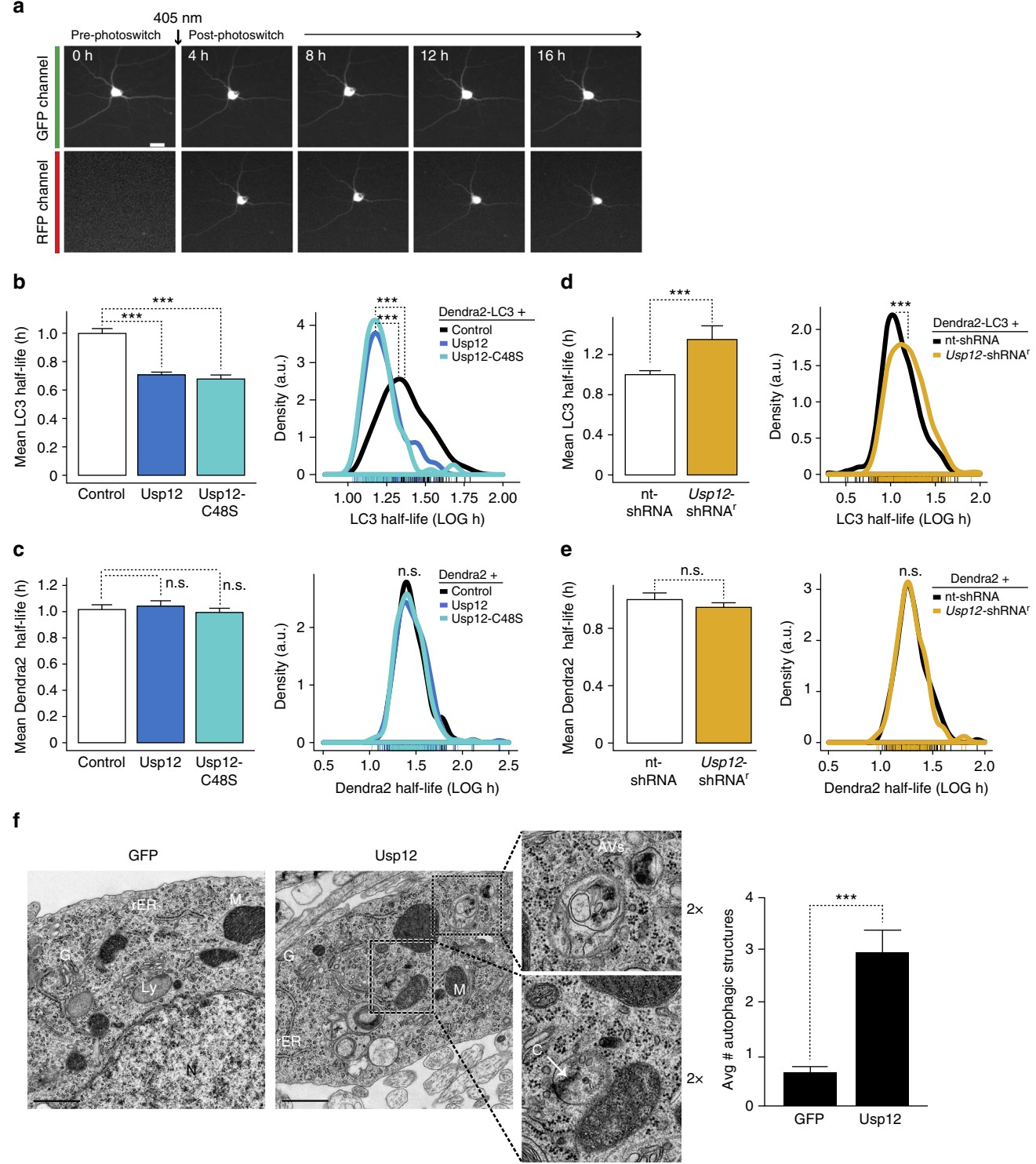

autophagic capacity, we conclude from these studies that Usp12 can function as a modulator of autophagic flux in primary neurons. Since a non-catalytic mutant of Usp12, Usp12-C48S, stimulated LC3 clearance as efficiently as Usp12, we additionally conclude that this function of Usp12 is independent of its deubiquitinating activity.

Interestingly, although these studies indicate endogenous Usp12 has a role in modulating autophagic flux, genetic knockdown of Usp12 is not overtly toxic to neurons. The level of reduction of autophagic flux by Usp12 knockdown under normal cellular conditions might not be sufficient to induce toxicity or other components of the autophagic pathway may sufficiently compensate for the loss of Usp12 function in the pathway under basal conditions. Under conditions of stress, however, such as with expression and accumulation of toxic mHTT protein, the loss of Usp12 function in regulating autophagic flux may contribute to the exacerbation of mHTT-associated neurotoxicity.

Despite the strong effect of Usp12 on autophagy, we did not see an effect of Usp12 on clearance of mHTT using a similar optical pulse-labeling assay to measure mHTT half-life. One possible

**Fig. 6** Usp12 modulates autophagy in neurons. **a** Representative images of an optical pulse-labeling experiment to measure Dendra2 or Dendra2-LC3 half-life in living neurons. Dendra2 protein is observed in neurons in the green fluorescence channel before and after 405 nm pulse photoswitch, but Dendra2 in the red fluorescence channel is only observed after the photoswitching pulse (scale bar = 20 μm). **b** Left panel, mean half-life of photoconverted Dendra2-LC3 in Usp12 or Usp12-C48S compared to control. Right panel, density plot showing distribution of Dendra2-LC3 half-lives in individual neurons expressing empty vector control, Usp12, or Usp12-C48S. **c** Usp12 or Usp12-C48S does not affect mean control protein Dendra2 half-life (left panel) or half-life distributions in individual neurons (right panel). **d** Mean half-life of Dendra2-LC3 (left panel) in *Usp12*-shRNA$^r$-expressing neurons compared to a non-targeting (nt) shRNA; (right panel) density plots. **e** Graphs of mean half-life (left) and density plots (right) of Dendra2 control in nt-shRNA or *Usp12*-shRNA$^r$ neurons. Number of neurons per group—Dendra2-LC3+: empty vector, n = 121; Usp12, n = 121; Usp12-C48S, n = 81; nt-shRNA, n = 341; *Usp12*-shRNA$^r$, n = 328; Dendra2+: empty vector, n = 162; Usp12, n = 210; Usp12-C48S, n = 173; nt-shRNA, n = 160; *Usp12*-shRNA$^r$, n = 130. p Value for mean half-life and distributions determined by Mann-Whitney and Kolmogorov-Smirnov tests, respectively (***p < 0.001). **f** Electron micrographs and quantification of autophagic structures in neurons transduced with Usp12 or GFP lentivirus. AV autophagic vacuoles, C curving phagophore, M mitochondria, G Golgi, rER rough endoplasmic reticulum, Ly lysosome, N nucleus. Quantification of autophagic structures represents the average number of autophagic vacuoles (pre-autophagic, autophagosome, and autolysosome structures) per EM image (×4000 micrograph). GFP, n = 28 images; Usp12, n = 33 images. Scale bar = 1 μm. Error bars represent s.e.m. (***p < 0.001). Scale bar = 1 μm

explanation is that a subpopulation of HTT conformers may exist that are especially toxic, and that Usp12 preferentially clears these. If that subpopulation of mHTT is small relative to the overall size of the mHTT pool, its clearance by Usp12 would elude detection with our optical pulse-labeling assay, which measures effects on all HTT species. Furthermore, if that conformer is not on the pathway that leads to IB formation, it would also explain why no effect of Usp12 on IB formation was observed. Another possibility is that Usp12 stimulation of autophagy confers neuroprotection primarily by clearing cargo other than mHTT. In support of this hypothesis, others have observed that the polyglutamine-expanded proteins can trigger a collapse in proteostasis that affects the levels of other metastable proteins[1] and have suggested that neurodegeneration may be the result of a complex loss of function of these other metastable proteins[45]. Toxic mHTT protein also interferes with cargo recognition that is required for efficient autophagy[46]. In this case, autophagosomes from HD neurons might inefficiently recognize and sequester cargo, such as cytosolic organelles and proteins, which in turn contribute to HD-associated cellular toxicity. Usp12 stimulation of autophagy, in the context of mHTT, could lead to more efficient cargo loading in autophagosomes, and thereby reduce mHTT-associated toxicity.

The specificity of Usp12 in modulating mHTT toxicity, compared to our neuronal models of PD and ALS, is of interest from a therapeutic perspective. Both α-synuclein and TDP-43 are substrates for autophagic degradation[33,47,48], and so, we might predict that stimulation of autophagy by Usp12 overexpression would also be neuroprotective in the related PD and ALS neuron models. However, we observed that Usp12 co-expression did not rescue neuronal toxicity caused by TDP-43 or α-synuclein expression. One potential explanation for this finding is that because polyQ-containing proteins interfere with the Beclin1-induced autophagy pathway[49], Usp12 expression may specifically compensate for mHTT-related defects in autophagy. Identification of signaling pathways that regulate Usp12 catalytic and non-catalytic functions will help to determine how Usp12 may be utilized as a neuroprotective agent for various disease models in which stimulation of autophagy could be beneficial. We have found an intriguing interaction of Usp12 with p62 and Optn, two key receptors necessary for the selective-autophagic clearance of misfolded proteins, mitochondria, and protein aggregates. Further studies will be needed to elucidate the relevance of these interactions for the selectivity of Usp12 in rescuing mHTT toxicity.

In summary, our studies identify Usp12 as a potent modifier of mHTT-mediated neurotoxicity, and a non-canonical deubiquitinase, which regulates autophagic flux in primary neurons. Our studies are consistent with previous publications indicating that upregulation of autophagy is neuroprotective in cell and animal models of HD[19,20].

Future studies directed toward understanding the precise molecular function of Usp12 in autophagy and its relationship to neurotoxicity will help to further establish the therapeutic potential of this genetic modifier for the treatment of HD.

## Methods

**Plasmids**. Human pGW1-HTT-N586 plasmids were generated by PCR amplifying the N-terminal 586 amino acids of full-length Q17 or Q138 human huntingtin, and inserted into pGW1-exon1-Q17-eGFP at *Kpn*I and *Bam*HI restriction sites. HTT-mEos2 plasmid was generated by replacing the enhanced GFP (eGFP) tag with the mEos2 sequence. TDP-43- and α-synuclein-bearing plasmids consisted of human TDP-43 and α-synuclein, respectively. The A315T mutation was created by site-directed mutagenesis of the adenine base at position 943 to guanine[27,28,33]. *Usp46* cDNA was amplified from HEK293 cell total RNA with primers 5′-TTT TTG GTA CCA TGA CTG TCC GAA ACA TCG CCT CC ATC-3′ and 5′-TTT TGG ATC CTT ACT CTC TTG ACT GAT AGA ATA AAA TAT ATC C-3′, and cloned into pGW1 plasmid at *Kpn*I and *Bam*HI restriction sites. Usp12-C48S, Usp12-C48A, Usp12-C50S, Usp12-DBLCS, and Usp46-C44S were generated by site-directed mutagenesis. Dendra2 was cloned into pGW1-CMV from pDendra2-N (Evrogen) using *Hin*dIII and *Mfe*I. pGW1-Dendra2-LC3 was generated by subcloning the LC3 fragment from pEGFP-LC3 into pGW1-Dendra2 plasmid[33,40]. shRNA oligos targeting rat *Usp12* 5′-TCG AGT ATA TCG ATG AAG CTG GTC CAT TGA TAT CCG TGG ACC AGC TTC ATC GAT ATA TTT TTT CCA ATT GG-3′), and nt-shRNA oligos (5′-CTC GAG GTA ATA TAG CAA GGC GAG AAT TCA AGA GAT TCT CGC CTT GCT ATA TTA CTT TTT TAA GCT T-3′) were cloned into pSilencer 1.0 or FUGW lentiviral vector.

**Cell culture and transfection**. Rat cortices were dissected from embryonic day 18–21 rats, digested with papain (10 U/ml) and 0.6 × 10⁶ cells/ml on a poly-lysine/laminin substrate[27,50]. Primary neuron cultures were transfected at 4–5 days in vitro with 50–600 ng plasmid DNA depending on the construct, and 0.5 μl of Lipofectamine 2000 per well in the 96-well plates. The incubation time of the Lipofectamine 2000/DNA complexes with the cells is limited to 30–60 min to prevent toxicity of the transfection to neurons. In general, transfection of primary neurons by lipofection results in 1–5% transfection efficiency, which sparsely labels the cultures and facilitates single-cell microscopy studies. HEK293T cells were maintained in Dulbecco's modified Eagle medium (DMEM)/10% fetal bovine serum, and PC12 cells were maintained in DMEM/10% horse serum, 5% fetal bovine serum. Cell lines were transfected with Lipofectamine 2000 according to the manufacturer's protocols (Life Technologies).

**Neuronal differentiation of NSCs**. Neural stem cells (NSCs) were maintained in Stemline (Sigma) supplemented with 100 ng/ml epidermal growth factor and fibroblast growth factor (Chemicon). Briefly, NSCs were differentiated according to the long protocol, whereby cells were incubated with NIM (DMEM:F12 supplemented with N2 and B27) for 5 days, followed by supplementation with 25 ng/ml brain-derived neurotrophic factor (BDNF; Peprotech) for 2 days, and then supplementation for 21 days with BDNF, 10 nM purmorphamine (Santa Cruz Biotechnology), and 100 ng/ml DKK (R&D), followed by supplementation with BDNF, 0.5 mM dibutyryl cyclic AMP (Sigma), and 0.5 mM valproic acid (Sigma)[24,31]. Transfections were performed with amaxa electroporation, according to manufacturer instructions at approximately day 28.

**Fly stocks**. Flies were raised on maize medium, in LD12:12 at 25 °C. The *elav-GAL4* [c155] and *actin-GAL4* [3954] drivers were obtained from the Bloomington Stock Center, Indiana. The *w;+;UAS Htt93Q exon 1*[25] flies were a gift from Larry Marsh and Leslie Thompson (University of California, Irvine). CG7023 RNAi lines

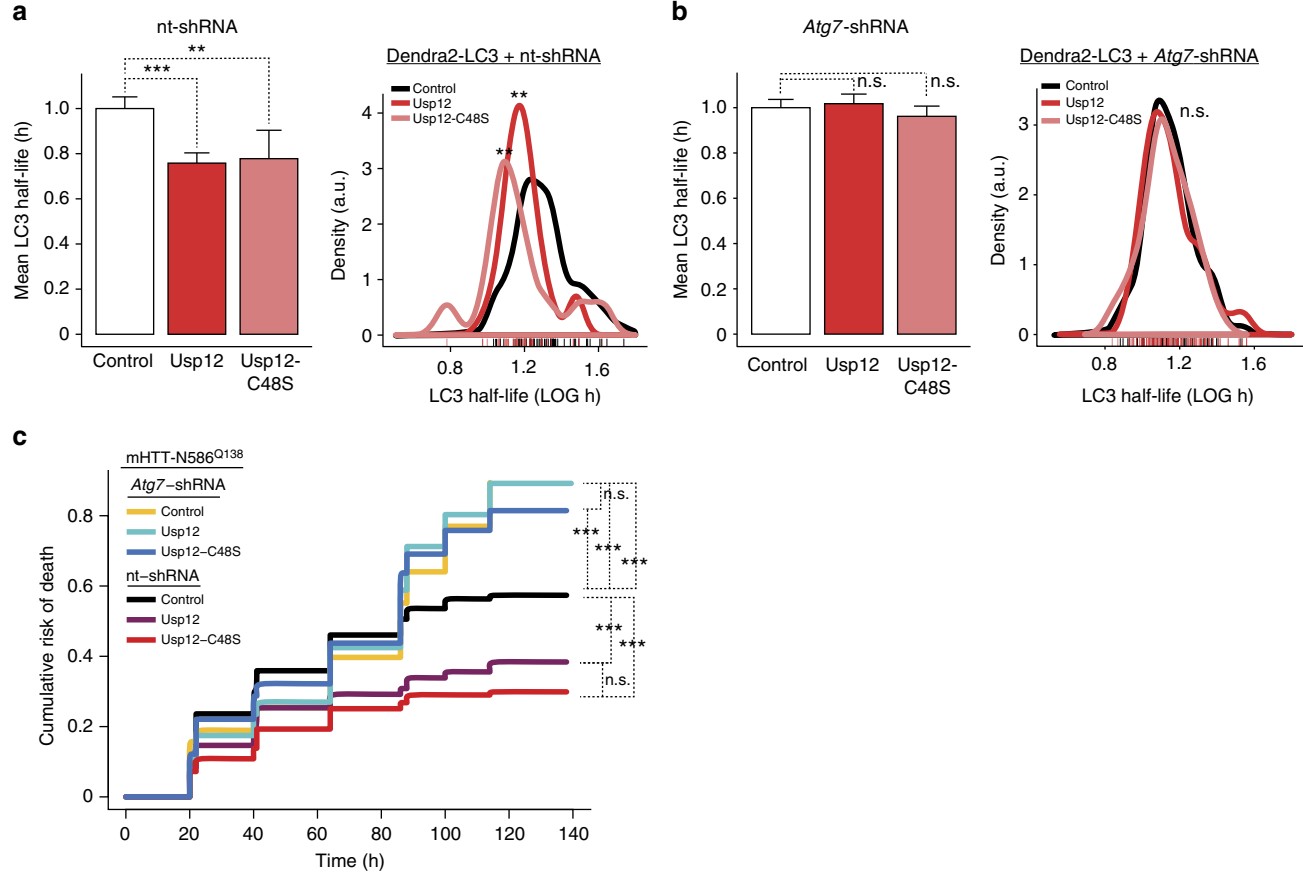

**Fig. 7** Usp12 function in LC3 turnover and mHTT-mediated toxicity is impaired in neurons genetically depleted of Atg7. **a**, **b** LC3 half-life was measured in neurons co-transfected with nt-shRNA or *Atg7*-shRNA, Usp12, Usp12-C48S, or control plasmids, and Dendra2-LC3 using the optical pulse-labeling assay described above. *P* value for mean half-life and distributions determined by Mann-Whitney and Kolmogorov-Smirnov tests, respectively. Error bars represent s.e.m. (**p < 0.01, ***p < 0.001). **c** Cumulative risk of death plots for neurons co-transfected with toxic mHTT and Usp12 overexpression or shRNA plasmids. Usp12 and Usp12-C48S do not effect neuroprotection during inhibition of Atg7-dependent autophagy. Cox proportional hazards analysis was used to estimate the relative risk of death, or hazard ratio (HR) and the *p* values were determined with the log rank test (***p < 0.001) (Supplementary Table 9)

were obtained from the Vienna Drosophila RNAi Center (stock # 108313). The UAS CG7023 overexpression flies were made by cloning the full-length CG7023 CDS from the BDGP clone RE52890 (Genbank BT010235) into the pJFRC2 vector[51], obtained from Addgene—plasmid # 26214) between the *Xho*I and *Xba*I sites. The resulting plasmid was purified and injected by BestGene Inc into the same attP insertion site as used by the VDRC (Landing site VIE-260B, VDRC stock # 60100). An additional control line was created by injecting the pKC26 vector without any hairpin construct into the VIE0260B background. The knockdown, control, and overexpression lines were crossed into the *w;+;UAS Htt93Q* genetic background using standard techniques, and the resulting flies crossed to the *elav-GAL4* driver to generate the experimental flies.

**Pseudopupil analysis**. The number of visible rhabdomeres per ommatidium was scored for 50–100 ommatidia per fly, with 8–10 flies examined per genotype at day 0, 1, or 7 post eclosion. Heads from aged flies were fixed to slides with fingernail polish, and rhabdomeres were examined at ×500 magnification using an Olympus BH2 microscope.

**Robotic imaging and analysis**. For survival analysis, neurons were imaged at approximately 24-h intervals after transfection for the number of days indicated in each experiment. Custom-based algorithms for ImageJ and Pipeline Pilot (Accelrys, San Diego, CA) software were used to track individual neurons based on fluorescence intensity of the transfected morphology marker. Survival or time of death was determined by a loss of morphology marker fluorescence, cell body rounding, or dissolution. Cox proportional hazards analysis was used to determine differences in survival among populations. Cumulative hazard plots were generated using custom-designed algorithms and the survival package within R. Optical pulse-labeling experiments were performed as follows: neurons were co-transfected with photoswitchable fluorescent protein plasmids (Dendra2 or Dendra2-LC3; HTT-mEos2) and potential modifier or control plasmids. Approximately 24–48 h after

transfection, neurons are pulsed with a 405-nm light for 4 s to photoconvert a proportion of the photoswitchable protein from green to red. Images are captured approximately every 4 h in green and red fluorescence channels for 3–5 time points, and then approximately every 24 h thereafter. Fluorescence in the red channel represents only the protein that existed at the time of photoconversion, while fluorescence in the green channel represents previously and newly synthesized protein. Intensity of the photoconverted protein in individual neurons is measured at each 4-h time point with the analysis software described above and used to calculate the protein half-life[33,34].

**Super-resolution microscopy**. Cells were seeded on glass-bottom gridded dishes (MatTek) and imaged with a Nikon N-SIM. Raw 3D-SIM images were acquired (15 images representing five phases and three rotations at each focal plane) typically using 200-ms exposure and Plan Apo IR ×60/1.27 objective. Images were reconstructed using NIS-Elements 4.12.

**Immunofluorescence**. At 48–72 h after transfection, neurons were treated with ammonium chloride (50 mM) or Bafilomycin A (25 nM) for 3–4 h and subjected to live-cell or fixed-cell imaging. Fixation occurred at room temperature with cold 4% paraformaldehyde for 10 min. Protein blocking was done with 5% goat serum/0.01% Triton in phosphate-buffered saline for 1 h at room temperature. Cells were incubated overnight with polyclonal rabbit anti-p62 (Abcam) and incubated with a secondary antibody Alexa-647 (1:500) for 1 h at room temperature.

**Protein expression immunoblotting and immunoprecipitation**. Protein lysates were prepared by standard methods. Briefly, cells were lysed in RIPA buffer (150 mM NaCl, 1% NP-40, 0.1% sodium dodecyl sulfate, 0.5% sodium deoxycholate, and 50 mM Tris, pH 7.4) supplemented with protease inhibitor cocktail (Roche). Bradford reagent or BCA protein assay kit was used to determine protein concentrations. Lysates were separated by SDS-polyacrylamide gel electrophoresis and

transferred to nitrocellulose for western analysis. Immunoprecipitations were performed with protein G Dynabeads according to the manufacturer's instructions (Life Technologies). Anti-Usp12 (Origene), anti-Usp46 (Atlas Antibodies), anti-GFP (Millipore), anti-tubulin (Abcam), anti-Optn (Bethyl labs), and anti-p62 (Progen) antibodies were used at 1:1000–1:5000 dilution. Fluorescent conjugated secondary antibodies (Li-Cor) were used at 1:10 000–20 000 dilutions. Signal was detected by scanning membranes with Li-Cor Odyssey CLx system. Quantification was performed with ImageStudio software (Li-Cor). Full immunoblots are showed in Supplementary Fig. 11.

**Lentivirus generation and EM.** HEK293T cells (Life Technologies) were transfected with Usp12-FUGW2 or GFP-FUGW, and delta8.9 and VSVG plasmids to generate lentivirus. Supernatant was collected 48–96 h after transfection and concentrated by PEG precipitation and centrifugation. Lentivirus was titered with Lenti-X p24 Rapid Titer Kit (Clontech). Primary rodent neuronal cultures transduced with GFP or Usp12 lentivirus for approximately 5 days, were fixed in 2% glutaraldehyde, 1% paraformaldehyde in 0.1 M sodium cacodylate buffer, pH 7.4, post-fixed in 2% osmium tetroxide in the same buffer, en block stained with 2% aqueous uranyl acetate, dehydrated in acetone, infiltrated, and embedded in LX-112 resin (Ladd Research Industries). Samples were ultrathin-sectioned on a Reichert Ultracut S ultramicrotome and counterstained with 0.8% lead citrate. Grids were examined on a JEOL JEM-1230 transmission electron microscope (JEOL USA, Peabody, MA) and photographed with the Gatan Ultrascan 1000 digital camera (Gatan, Warrendale, PA).

**Bioinformatics and statistical analysis.** Protein interaction network maps were generated with the open source software for network visualization, Cytoscape (version 3.2.1)[52]. Protein interactions were imported from BIOGRID, a general repository for interaction datasets[53]. Networks for HTT, Usp12, Wdr20, and Optn were visualized and merged to generate the interaction map, with each of these proteins representing a node or hub in the network. Interactors were filtered to include only first neighbors of each of the nodes. Protein IDs (abbreviations) for each interactor are indicated on the network map in the Supplementary Fig. 7. Cox proportional hazards analysis was used for survival experiments. For half-life experiments, half-life values were determined by first-order exponential decay modeling, mean half-lives were compared by Mann-Whitney Wilcoxon test, and distributions compared with two-sided Kolmogorov-Smirnov test. Mean protein expression levels from western blot and EM analysis were compared with Student's *t*-test.

**Quantitative polymerase chain reaction analyses from mammalian cells.** Total RNA of cells was prepared with RNeasy Micro Kit (Qiagen) in accordance with the manufacturer's instructions. Single-stranded complementary DNA was produced by reverse transcription using 1 µg of DNA-free RNA in a 20-µl reaction High-Capacity cDNA Reverse Transcription Kit (Applied Biosystems). Quantitative polymerase chain reaction (qPCR) was performed using the TaqMan probe-based system (Applied Biosystems) on the ABI 7900HT as per the manufacturer's instructions (Applied Biosystems; Wdr20 Rn01430046_m1; Wdr48 Rn01447049_m1; *beta-Actin* Rn00667869; *Usp12* Rn01774390_g1).

**qPCR analyses from *Drosophila* samples.** For qPCR quantification of relative *Usp12* expression in *Drosophila*, CG7023 RNAi and UAS CG7023 overexpression lines were crossed to the *actin*-GAL4 driver. Newly emerged adult offspring (day 0) were snap-frozen in liquid nitrogen and stored at −80 °C before RNA extraction. Six biological replicates consisting of 15 whole fly bodies each were used for the analysis groups. RNA was extracted using TRIzol reagent (Ambion) according to the manufacturer's instructions. RNA was treated with TURBO DNase (Ambion) to remove any traces of genomic DNA, before synthesis of cDNA using the QuantiTect Reverse Transcription Kit (Qiagen). CG7023 and *rp49* primer sequences were obtained through Fly PrimerBank[54]. Primers were tested for specificity and optimum annealing temperature using temperature-gradient PCR prior to qPCR, and melt curve analysis was performed on qPCR products. qPCR reactions were performed on a LightCycler 480 system (Roche) using Maxima SYBR Green master mix (ThermoFisher Scientific). Total reaction volume was 10 µl, with forward and reverse primer concentrations of 330 nM. Four technical replicates were used for each sample and a control reaction in which no reverse transcription was carried out was also included. Crossing points (Cp) were determined by the second derivative method using LightCycler 480 Software (Roche). For relative expression quantification, raw fluorescence data of technical replicates were averaged for each sample. The amplification efficiency of each reaction was calculated using the qpcR package in R Studio[55] by fitting sigmoidal curves to the raw fluorescence data, using the *pcrbatch* function. The relative expression of CG7023 in overexpression and RNAi lines was normalized to each corresponding UAS control line, calculated using the *ratiobatch* function with the Cp value of *rp49* used as an internal control for each sample. Statistical significance of relative expression levels was tested using a pairwise-reallocation test based upon that used by REST software[56], where Cp and efficiency values were permuted within control and treatment groups.

**Data availability.** The data that support the findings of this study are available from the corresponding author upon reasonable request.

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

## Acknowledgements
We thank members of the Finkbeiner Laboratory for helpful suggestions and comments; Gary Howard and Eric Martens for editorial assistance; and Kelley Nelson for administrative assistance. We thank Larry Marsh and Leslie Thompson for providing the *w;+; UAS Htt93Q exon 1 Drosophila melanogaster* strain. This work was done with support from NIH NS039074, U54 HG008105, and NS R37101996; the Taube/Koret Center; and the Gladstone Institutes (S.F.). E.W.G. was supported by a research contract from the CHDI Foundation awarded to F.G., while D.C.M. was supported by a Ph.D. studentship from the Midlands Integrative Biosciences Training Partnership funded by the BBSRC. We thank the Hereditary Disease Foundation for fellowship support to R.A.

## Author contributions
R.A. and S.F. developed the concepts for the manuscript and designed the experiments. R.A., P.P., and K.O.-N. performed the experiments on mammalian cells and analyzed the data. J.S.W. performed the electron microscopy analysis. E.W.G., D.C.M., and F.G. designed the *Drosophila* studies. E.W.G. and D.C.M. performed the experiments and discussed results in the *Drosophila* model. A.C.D. contributed reagents. R.A., P.P., and S. F. prepared the manuscript and all authors contributed to editing the paper. A. O. O. provided technical assistance. P. M contributed to the conceptual inception of the project and the initial stages of its development.

## Additional information

**Competing interests:** The authors declare no competing interests.

