## [Peer Review File · Nature Communications]

Editorial Note: this manuscript has been previously reviewed at another journal that is not operating a transparent peer review scheme. This document only contains reviewer comments and rebuttal letters for versions considered at *Nature Communications*.

REVIEWERS' COMMENTS:

Reviewer #1 (Remarks to the Author):

The authors have successfully addressed my concerns raised in the original review of the manuscript. The only outstanding issue is there appears to be some issue with the images presented in supplemental figure 7b. In my version, there are large white boxes occluding a juxtannuclear region of the cell. It is unclear if those boxes are present to serve a purpose but there is no indication of their purpose in the figure legend. This will need to be addressed prior to publication.

Reviewer #2 (Remarks to the Author):

The manuscript by Aron presents some interesting findings. They show that overexpression of dequiquinase Usp12 can reduce mHTT-mediated neurodegeneration in HD neuronal cell models and *Drosophila*. They found that Usp12 is a potent inducer of neuronal by accelerating autophagic flux and inducing an increase in autophagic structures.

However, the studies do not seem to be able to address two important issues. One is the selectivity of Usp12 on mHTT. Usp12 plays a role in regulating the stability or turn-over of a large number of proteins. It remains unclear why it can specifically target mHTT. The other issue is that many data were generated in vitro. It is known that mHTT toxicity is critically dependent on its expression level. There is no strong evidence that Usp12 has protective effect against mHTT at the endogenous level. The authors used human neuronal cells that were derived from HD iPCS. However, these cells do not show the accumulation of mHTT. It remains unknown whether these cells can really mimic neuropathology in the HD brain that requires much longer time to occur. It would be important for the authors to use HD knock-in mice to provide solid evidence to support the conclusion of this paper. Also, the mechanistic studies can be strengthened to give one a clear idea how autophagic structures are induced by Usp12.

Point by point response to reviewers

Reviewer #1 (Remarks to the Author):

The authors have successfully addressed my concerns raised in the original review of the manuscript. The only outstanding issue is there appears to be some issue with the images presented in supplemental figure 7b. In my version, there are large white boxes occluding a juxtannuclear region of the cell. It is unclear if those boxes are present to serve a purpose but there is no indication of their purpose in the figure legend. This will need to be addressed prior to publication.

We apologize for this issue related to the formatting of the supplemental figure 7b, and thank the reviewer for calling it to our attention. The figure has been corrected.

Reviewer #2 (Remarks to the Author):

The manuscript by Aron presents some interesting findings. They show that overexpression of dequiquinase Usp12 can reduce mHTT-mediated neurodegeneration in HD neuronal cell models and Drosophila. They found that Usp12 is a potent inducer of neuronal by accelerating autophagic flux and inducing an increase in autophagic structures.

However, the studies do not seem to be able to address two important issues. One is the selectivity of Usp12 on mHTT. Usp12 plays a role in regulating the stability or turn-over of a large number of proteins. It remains unclear why it can specifically target mHTT. The other issue is that many data were generated in vitro. It is known that mHTT toxicity is critically dependent on its expression level. There is no strong evidence that Usp12 has protective effect against mHTT at the endogenous level. The authors used human neuronal cells that were derived from HD iPSCs. However, these cells do not show the accumulation of mHTT. It remains unknown whether these cells can really mimic neuropathology in the HD brain that requires much longer time to occur. It would be important for the authors to use HD knock-in mice to provide solid evidence to support the conclusion of this paper. Also, the mechanistic studies can be strengthened to give one a clear idea how autophagic structures are induced by Usp12.

We thank the reviewer for the insightful critique of the study, and we do intend to further investigate these open questions. Given the novelty of Usp12 as new player in neuronal autophagy, our future work may go in multiple directions to deeply investigate (1) The specificity of Usp12 to rescue mHTT toxicity (2) The mechanism by which Usp12 targets mHTT clearance either directly or indirectly (3) How Usp12-mediated autophagy rescues mHTT toxicity and (4) the downstream substrates of catalytic vs. noncatalytic Usp12 in the context of protein aggregate clearance.

A potential explanation for the selectivity of Usp12 for HTT may reside in the context-specificity of the autophagy pathway in presence of polyQ stretches like that in the mHTT protein. A recent study has demonstrated that polyQ competes for binding of Beclin-1 for autophagosome formation and interferes with neuronal autophagy (Nature. 2017 May 4;545(7652):108-111). It is possible that Usp12 is acting upstream in the autophagy machinery and compensating for the loss of Beclin1 function in polyQ-expressing neurons. In this study, we have demonstrated that Usp12 interacts with specific autophagy receptors (p62 and Optineurin) which could be important for the selective clearance of toxic mHTT oligomers as opposed with other disease-specific aggregates. We have extended the discussion and included these comments, hoping to raise interest and discussion in the scientific community.

Concerning the choice of experimental system, there is no perfect disease model of mutant Huntingtin toxicity, and the ability of each model to recapitulate human disease pathogenesis is under debate given the extremely high variability of pathological marks observed in patients. In order to address this concern and gain confidence that our findings might be relevant to the human disease process, we used several complementary models of the HD. The overexpression of an N-terminal exon 1 fragment of mHTT in vivo is the most widely used model of Huntington disease, given its ability to recapitulate several disease-related hallmarks in humans, and we see effects of Usp12 in vivo in such a model. However, one limitation of this model is that exon1 lacks regulatory regions of mHTT that mediate clearance. Indeed, it has been demonstrated that conserved serine residue (S421) is important for mHTT conformation and clearance (J. Clin. Invest. 2016 Aug 15;126, 3585–3597). Based on this rationale, we used an N-terminal HTT fragment that encompasses this residue and has been shown to reproduce HD-related phenotypes in vivo, including brain atrophy, hyperactivity, abundance of HTT polypeptides and early mortality (J Neurosci. 2012 Jan 4;32(1):183-93). Finally, we sought to overcome the limitations of the overexpression model by validating our findings at endogenous levels of mHTT using a patient-derived iPSCs that were previously shown to recapitulate several key aspects of the disease (Cell Stem Cell. 2012 Aug 3;11(2):264-78). Since we had the same conclusions using three different models of the disease, we are confident that these findings are not an artifact of a particular disease model.

In sum, we report for the first time a novel role for Usp12 modulating mHTT toxicity *in vitro* and *in vivo* in a variety of models and demonstrate through a rigorous set of experiments that Usp12 potently engages and upregulates the autophagy / lysosomal pathway. Many of the suggestions made by the reviewer are ones we want to pursue in the future to build on the substantial body of work described in this manuscript but we respectfully argue are beyond the scope of the current study.